# OpenReview forum: "Mind the Gap: Evaluating Model- and Agentic-Level Vulnerabilities in LLMs with Action Graphs"
_ICLR.cc/2026/Workshop/AFAA — AFAA 2026 Poster_

### Official Review · Reviewer_84JP · 2026-02-12
**Mind the Gap: Evaluating Model- and Agentic-Level Vulnerabilities in LLMs with Action Graphs**

**Rating:** 3
**Confidence:** 4

**Summary:**

The paper describes AgentSeer - a framework used to evaluate agentic systems and illustrate how agent risks are different from model-level risks. It demonstrates that tool calling introduces vulnerabilities to the system and how it can amplify risk. They also provide a few techniques to mitigate such risks. They evaluate on GPT-OSS-20B and Gemini-2.0-flash using HarmBench objectives and PAIR-style iterative attacks.

**Strengths:**

The following are the strengths of the paper:
1) Through clear experimentation the authors demonstrate how model level risks are different from agentic level risks. They also show how tool calling can introduce vulnerabilities into the system.
2) Breaking down the system into a graph based on tools, prompts, responses and actions is extremely meaningful and can demonstrate logical flows to the user.
The following findings are noteable:
1) Iterative attacks reveal “agentic-only” vulnerabilities
2) Action-level hardening significantly reduces ASR

The authors have also showcased this with rigorous experimentation and by also using both open-source and non-open source models. This shows that the basis is solid and this is a real world problem that exists today.

**Weaknesses:**

The major concern with this paper is how it fits with the theme of this workshop - the workshop is of algorithmic fairness and this shows more promise as a security based paper. This paper does not echo any themes of fairness, bias or societal disparity.
That being stated, the following are the other areas that can be addressed:
1) GPT-4o serves as the judge for determining an attack and also for prompt hardening. I feel some bias may be introduced by this. Having distinct models serve for the different tasks might have been useful here. Also would have been interesting to see how different models behave as judges.
2) This system also evaluates a single architectural setting, which questions the generalizability of the paper. The scope is limited to a single domain and also the system architecture remains unchanged.

---

### Official Review · Reviewer_CXn4 · 2026-02-18
**AgentSeer is a valuable research direction in breaking down Agentic AI tasks into action-component graphs and evaluating them in adversarial settings**

**Rating:** 4
**Confidence:** 4

**Summary:**

The proposed framework AgentSeer decomposes agentic executions into granular action–component graphs and provides a standardized methodology and empirical validation of Attack Success Rate (ASR) for safety of agentic systems.

By giving complete execution observability, utilizing jail break attacks across open source agentic models in GPT-OSS-20B and Gemini-2.0-flash models, this work aligns to the ICLR 2026 AFAA Workshop goals.

**Strengths:**

1. The paper is well structured, flows across sections seamlessly with a good motivation and related work.

2. A thorough evaluation mechanism using Attack Success Rate (ASR) is presented across 2 models - GPT-OSS-20B and Gemini-2.0-flash on individual actions with GPT-4o-mini serving as a judge.

3. HarmBench is used on direct and iterative jail break attacks to isolate only agentic vulnerabilities. Further, an Automatic Prompt Hardening method reduces jail break success and demonstrates cross-model transfer.

4. Framework and results are presented through easy to understand figures, graphs and tables.

**Weaknesses:**

A. Major Concerns
1. Why was just the Shopping domain selected as the task? Why not other domains and since there is no other domain available for comparison, are the limitations enough to defend the claims presented throughout the paper?

2. Once the setup was finalized, Ablation runs could have fortified the experimental results.

3. Since, this study is focused only on one domain of Shopping and evaluates only using ASR, its rationale should be explained and also  not using any other metrics for example, Refusal Rate, Pass@k (completion rate), Tool Correctness, Plan Adherence, etc. should also be considered for evaluation. And, if not used then the reason should be mentioned as well.

4. Can the scatter plot analysis of ASR vs input token length be combined into a single plot across the models? That way the graphs become condensed to a single page and also one gets to see if there's any correlation / overlap in the common actions (despite the slight difference in numbers) across models? Can this be extended to other graphs as well?

5. Code is not shared for reproducibility and transparency.


B. Minor Improvements
1. The full form Attack Success Rate (ASR) should be mentioned in section 4.1 at its first usage after the Abstract section instead of the current section 4.2.

2. On Page 12, try condensing the brackets to the previous page by incorporating tab spaces on the same line so that you don't get a near empty page.

3. On Page 16, think of removing redundancy in the caption 'Tool Risk per Tool' by only keeping a single word 'Tool' as it is already understood. Same applies to 'Agent Risk per Agent'on Page 18.

---

### Official Review · Reviewer_ahji · 2026-02-24
**Review: MIND THE GAP: EVALUATING MODEL- AND AGENTICLEVEL VULNERABILITIES IN LLMS**

**Rating:** 4
**Confidence:** 4

**Summary:**

This paper introduces AgentSeer, an observability-based framework for evaluating the safety of agentic LLM systems, and uses it to show the gap between model-level and agentic-level  vulnerability. The authors construct an agentic testbed, decompose its execution into  actions and components and then  red-team two models (GPT-OSS-20B, Gemini-2.0-flash) using HarmBench objectives  They show that agentic contexts surface "agentic-only" vulnerabilities—objectives that fail at the model level but succeed when tool-calling, multi-step reasoning etc. are involved. The paper further proposes an automated, action-level prompt hardening loop that reduces average attack success rates substantially  and demonstrates partial cross-model transfer of the hardened prompts.

**Strengths:**

1. **Well-motivated problem framing.** The core thesis—that model-level red-teaming misses deployment-time risk arising from agentic use—is clearly articulated and supported with quantitative evidence. The 24–60% higher ASR in tool-calling contexts is a compelling data point.
2. **Systematic evaluation methodology.** The three-tier attack design (model-level iterative, agentic direct transfer, agentic iterative PAIR) provides a clean experimental ladder that isolates the contribution of agentic context to vulnerability.
3. **Actionable defense mechanism.** The action-level prompt hardening loop is a practical contribution: mine jailbreaks per action, attribute them , iteratively improve SIs and validate via replay. The large ASR reductions are encouraging.
4. **Cross-model validation.** Evaluating two architecturally distinct models (open-source vs. proprietary) with different safety-training profiles strengthens the generality of the structural findings

**Weaknesses:**

1. **Single-testbed, single-domain evaluation.** All experiments use one 6-agent Shopify analyst architecture built on LangGraph. It is unclear whether the observed vulnerability patterns  generalize to other agentic topologies domains, or frameworks. Even one additional testbed would significantly strengthen the claims. The authors acknowledge this in limitations
2. **Test models uses**: Both the test models used are not close to State of the art agentic models (e.g. Opus 4.6, GPT 5.3 etc) and are 1-2 generations old. It is unclear if the observed patterns will still apply to larger more sophisticated models that are more often deployed in agentic steups
3. **No utility-regression measurement.** The prompt hardening reduces ASR but the paper does not report whether the hardened prompts degrade task performance on benign inputs.
4. **LLM-as-judge reliability.** GPT-4o-mini serves as both the prompt-improvement model and the jailbreak judge, and only a binary 10/10 threshold is used. No inter-annotator agreement, human validation sample, or sensitivity analysis on the threshold is provided.

---

### Meta-Review · Area_Chair_4YVE · 2026-02-27

**Recommendation:** Main Papers Track
**Confidence:** 5

**Metareview:**

This paper demonstrates that agentic AI systems have vulnerabilities and can amplify risk in a way non-agentic systems don't.  This paper introduces AgentSeer, an observability-based framework for evaluating the safety of agentic LLM systems, and uses it to show the gap between model-level and agentic-level vulnerability.

Though the results are limited and in seemingly early stages, this is an interesting result that should yield interesting and generative discussion at the workshop. The authors should be sure to address the points raised by the reviewers.

---

### Decision · Program_Chairs · 2026-03-02

Accept (Poster)